# MRP8/14 Is a Molecular Signature Triggered by Dopamine in HIV Latent Myeloid Targets That Increases HIV Transcription and Distinguishes HIV+ Methamphetamine Users with Detectable CSF Viral Load and Brain Pathology

**DOI:** 10.3390/v15061363

**Published:** 2023-06-13

**Authors:** Liana V. Basova, Alexander Lindsey, Annemarie McGovern, Ashley Rosander, Violaine Delorme-Walker, Wael M. ElShamy, Ved Vasishtha Pendyala, Peter Jesse Gaskill, Ronald J. Ellis, Mariana Cherner, Jennifer E. Iudicello, Maria Cecilia Garibaldi Marcondes

**Affiliations:** 1San Diego Biomedical Research Institute, San Diego, CA 92121, USA; 2Human Biology Program BISP, University of California San Diego, San Diego, CA 92037, USA; 3Systems Biology Internship, Creighton Preparatory School, Omaha, NE 68114, USA; 4College of Medicine, Drexel University, Philadelphia, PA 19102, USA; 5HIV Neurobehavioral Research Program, University of California San Diego, San Diego, CA 92103, USA

**Keywords:** dopamine, biomarkers, methamphetamine, HIV

## Abstract

There is a significant overlap between HIV infection and substance-use disorders. Dopamine (DA) is the most abundantly upregulated neurotransmitter in methamphetamine abuse, with receptors (DRD1-5) that are expressed by neurons as well as by a large diversity of cell types, including innate immune cells that are the targets of HIV infection, making them responsive to the hyperdopaminergic environment that is characteristic of stimulant drugs. Therefore, the presence of high levels of dopamine may affect the pathogenesis of HIV, particularly in the brain. The stimulation of HIV latently infected U1 promonocytes with DA significantly increased viral p24 levels in the supernatant at 24 h, suggesting effects on activation and replication. Using selective agonists to different DRDs, we found that DRD1 played a major role in activating viral transcription, followed by DRD4, which increased p24 with a slower kinetic rate compared to DRD1. Transcriptome and systems biology analyses led to the identification of a cluster of genes responsive to DA, where S100A8 and S100A9 were most significantly correlated with the early increase in p24 levels following DA stimulation. Conversely, DA increased the expression of these genes’ transcripts at the protein level, MRP8 and MRP14, respectively, which form a complex also known as calprotectin. Interestingly, MRP8/14 was able to stimulate HIV transcription in latent U1 cells, and this occurred via binding of the complex to the receptor for an advanced glycosylation end-product (RAGE). Using selective agonists, both DRD1 and DRD4 increased MRP8/14 on the surface, in the cytoplasm, as well as secreted in the supernatants. On the other hand, while DRD1/5 did not affect the expression of RAGE, DRD4 stimulation caused its downregulation, offering a mechanism for the delayed effect via DRD4 on the p24 increase. To cross-validate MRP8/14 as a DA signature with a biomarker value, we tested its expression in HIV+ Meth users’ postmortem brain specimens and peripheral cells. MRP8/14+ cells were more frequently identified in mesolimbic areas such as the basal ganglia of HIV+ Meth+ cases compared to HIV+ non-Meth users or to controls. Likewise, MRP8/14+ CD11b+ monocytes were more frequent in HIV+ Meth users, particularly in specimens from participants with a detectable viral load in the CSF. Overall, our results suggest that the MRP8 and MRP14 complex may serve as a signature to distinguish subjects using addictive substances in the context of HIV, and that this may play a role in aggravating HIV pathology by promoting viral replication in people with HIV who use Meth.

## 1. Introduction

After entering target cells, HIV reverse-transcribes its dimeric RNA genome into a complementary DNA and a double-stranded DNA provirus that integrates into the host genome, utilizing host machinery to activate its genome or remaining in a latent state [1,2,3]. This is remarkably important in reservoir organs such as the central nervous system (CNS), where the virus enters early and infects innate immune, glial, and infiltrating myeloid cells. While latent, the virus is rarely targeted by the immune response. Replication may make the virus susceptible to antiretroviral therapy (ART) effects, but may also be associated with inflammation and aggravated neurological disorders. On the other hand, the mechanisms that maintain latency and low levels of inflammation are poorly understood, especially in the CNS. Understanding these mechanisms is particularly important in the context of substance-use disorders, particularly methamphetamine (Meth), where higher cerebrospinal fluid (CSF) and brain viral loads are observed compared to non-Meth users [4]. In humans, less-than-perfect ART adherence may contribute to poorer outcomes in drug users [5]. When ART is interrupted in patients who initiated therapy during the chronic phase of infection, with undetectable or low viral persistency, replication resumes [6,7,8,9,10,11], risking disease progression and the negative consequences of increased inflammation. However, addictive substances may trigger molecular mechanisms that act on the susceptibility to infection, viral replication, or latency.

HIV latency is a challenge for cures [12,13] and is a special problem in the CNS, where HIV target cells are mostly myeloid [12,13,14,15,16,17,18,19,20], rather than lymphoid, as in most other tissues. Whether substance-use disorders, including Meth, affect the reservoir status of persistently infected cells to promote viral replication has not been fully examined.

Meth and other stimulants cause a significant increase in specific neurotransmitters in the brain, with dopamine (DA) being the most abundantly upregulated. Meth disrupts DA reuptake and packaging through effects on the DA transporter (DAT) [21,22,23,24,25] and the vesicular monoamine transporter 2 (VMAT2) [25,26,27], increasing cytoplasmic DA in neurons by blocking reverse-transport and attenuating clearance [28,29,30,31,32,33]. Combined with HIV infection, dopaminergic disorders are further enhanced because the expression of DA receptors (DRDs) is altered by the virus or its proteins [34,35,36,37,38]. The imbalance between DA levels and dopaminergic circuits underlies reward and conditioning, contributing to the compulsive drug use and loss of control in addiction [39]. Yet, the effects of DA go beyond neuronal circuits. DA receptors (DRD1-5) are expressed not only by neurons, but also by a diversity of cell types, including myeloid cells that are the targets of HIV in the brain and elsewhere [40,41,42,43,44]. This makes these cells responsive to the hyperdopaminergic environment that is characteristic of stimulant drugs such as Meth [45,46,47,48,49]. Importantly, DRD levels and their regulatory components can be modified by HIV, as well as Meth [34]. For instance, HIV proteins such as Tat can directly modify the DA system, both by decreasing the expression of DRD1 and DRD2 [34], and by affecting the DA transporter (DAT) [50]. Conversely, the HIV pathogenesis is singular in the context of drug abuse, particularly Meth, as it is influenced by immune signatures triggered by the hyperdopaminergic environment and interactions with the infection in the CNS.

Our lab has previously shown that DA is involved in the increase in the brain viral load in the context of substance use by upregulating the coreceptor for HIV entry of CCR5 on myeloid cells via DRD1 signaling, thus increasing the susceptibility to viral spread [44]. Changes in DA levels have been suggested as a factor impacting HIV infection in a CCR5-dependent manner by us and others [42,43]. The effect of DA on viral latency as a factor increasing the viral load in the brain has not been examined. The characteristics of brain reservoir cells and the signatures that are associated with latency, active replication, or the induction of reverse latency in drug abusers are unknown.

We tested the hypothesis that HIV+ Meth users may have an increased CNS viral load due to effects of DA on myeloid cell latency. Using U1 cells as a model of latent myeloid infection, we defined the effects of DA, established DRDs that selectively signal or prevent latency reversal, and identified myeloid biomarkers associated with these phenotypes. The DA signatures associated with latency reversal included the onset of a subset expressing S100A8 and S100A9 gene transcripts, corresponding to the heterodimers of the S100 family of the Ca^2+^-binding-proteins complex, MRP8 and MRP14, respectively, which form a complex known as calprotectin. Latent U1 cells also expressed the MRP8/14 receptor, an advanced glycation-end products (AGE) receptor (RAGE). We confirmed the ability of MRP8/14 to reverse latency via RAGE, followed by the examination of whether these molecular characteristics can be detected in the brain and blood of human HIV+ subjects that are Meth users in correlation with uncontrolled viral load in the cerebrospinal fluid.

The results suggest novel underlining mechanisms associated with HIV latency in myeloid cells, which are disrupted by DA in the context of Meth, providing an explanation for uncontrolled brain and CSF viral loads in drug users and a potential biomarker.

## 2. Materials and Methods

**Cell lines**—Chronically infected HIV-1 promonocytic (U1, ARP-165) cell lines were originally derived by limiting the dilution cloning of U937 surviving an acute infection with HIV-1 by Folks et al. [51]. The cells were obtained through the NIH HIV Reagent Program, Division of AIDS, NIAID, NIH: antihuman CD34 hybridoma (PR18) and ARP-165, contributed by Dr. Thomas Folks. The U1 cells were cultured in RPMI 1640 containing 2.0 mM of L-glutamine and 10% heat-inactivated fetal bovine serum (Lonza Bioscience, Morrisville, NC, USA), and maintained in the log phase with >98% viability prior to plating at 2 × 10^6^/mL in 24-well plates and stimulation.

**Treatments in vitro**—All concentrations were optimized prior to experimentation. Dopamine hydrochloride (H8502, Sigma Aldrich, St. Louis, MO, USA) was added at 1 and 10 μM. Latency reversal agents were Ingenol-3-angelate (PEP005) at 0.1 and 10 μM (Sigma-Aldrich, St. Louis, MO, USA), bromodomain and extraterminal domain inhibitor 151 (BETi) at 1 and 10 μM (Tocris, Minneapolis, MN, USA), and phorbol myristate acetate (PMA) at 10 and 100 ng/mL (Sigma-Aldrich). Selective DA receptor agonists SKF38393 (DRD1/5), ropinirole (DRD2), pramipexole (DRD3), and PD168077 (DRD4) (all from Tocris) were added at 1 and 10 μM. Selective DRD antagonists SCH23390 (DRD1/5), haloperidol hydrochloride (DRD2), PNU177864 (DRD3), and L745870 hydrochloride (DRD4) (all from Tocris) were added at 1 μM to cultures, alone or combined with 1 and 10 μM DA. Recombinant human heterodimer MRP8/14 proteins (S100A8/S100A9, R&D Systems, Minneapolis, MN, USA) were used at 0.5, 1, and 5 μM. The RAGE antagonist FPS-ZM1 (Tocris) was used alone or simultaneously with other stimuli at 1 and 5 μM. All stimulations, with single or combined compounds, were tested at 24 h.

**Quantification of HIV replication**—HIV p24 was measured in the supernatants 24 h after stimulation using an ELISA kit (BioXpress International, Frederick, MD, USA), and viral RNA was measured by a digital droplet PCR using normalized msTatRev and skGag primers, performed at the University of California San Diego Translational Virology Core.

**Detection of MRP8/14 in culture supernatants**—Culture supernatants were collected after stimulation, and levels of MRP8/14 were measured using the LEGENDplex 8.0 microbeads immunoassay (Biolegend, San Diego, CA, USA) by flow cytometry [52,53,54] using human MRP8/14 capture beads A5 13X, following the manufacturer’s instructions, acquired in a CytoFlex S (Beckman Coulter, Brea, CA, USA) and analyzed in the cloud-based LEGENDplex software version 8 (Biolegend).

**Detection of MRP8/14 by ELISA in protein preparations**—Following stimulation, the cells were collected from tissue culture plates, washed once with ice-cold PBS, and then homogenized in ice-cold RIPA buffer containing 1 mM protease inhibitor PMSF and 10% complete phosphatase inhibitor (Roche, Indianapolis, IN, USA). The protein concentration in the lysate was determined with the BCA assay (Pierce Biotechnology, Waltham, MA, USA). Cytosolic and membrane fractions were separated with the MEM-PER Plus kit (ThermoFisher, Waltham, MA, USA). MRP8/14 levels were tested using the Legend Max ELISA kit (BioLegend) following the protocol from the manufacturer.

**Transcriptional profiles**—The total cellular RNA was extracted from the U1 cell cultures at a density of 10^6^/mL, 12 h following exposure to DA or selective agonists. Extraction was performed using the RNAasey Mini Kit (Qiagen, Germantown, MD, USA) according to the manufacturer’s protocol. Total RNA was quantified by measuring the total absorbance at OD260/OD280 nm in Nanodrop (Thermo Fisher Scientific, Watham, MA, USA), and the quality of the total RNA was monitored using an Agilent 2100 Bioanalyzer (Agilent Technologies, Santa Clara, CA, USA).

Transcriptional changes were examined in biological triplicates of DA, DRD1/5, and DRD4 agonists using the Agilent Whole Human genome platform, performed by Phalanx Biotech (San Diego, CA, USA). Normalized values were compared using the log fold change between conditions, with an alpha = 0.05.

**Systems biology and visualization**—Transcriptional changes caused by DA and DRD agonists, compared to vehicle controls, were visualized using Genemania [55,56] in the Cytoscape 3.3 interface [57,58], with links through the Pathway Commons algorithm (blue lines) and public databases of shared protein domains (Red lines). Blue shapes represent downregulation and yellow shapes represent upregulation. Circles represent *p* < 0.05 compared to the vehicle.

**Human cohorts**—Participants included 102 adults enrolled by the University of California San Diego’s HIV Neurobehavioral Research Program (HNRP) and Translational Methamphetamine Research Center (TMARC) under informed consent and approved protocols. Cognitive scores were collected using previously described protocols [59]. Peripheral blood leukocytes were collected during visits and archived. The subjects selected for this study were males between 35 and 49 years old due to characteristics of the cohort and for decreasing confounders. The participants were divided based on HIV serostatus (HIV+/−) and on Meth use (METH+/−). METH+ was defined as having met the lifetime DSM-IV criteria for methamphetamine abuse or dependence. A cross-sectional design assembled the following groups: HIV-METH− (*n* = 27), HIV+METH− (*n* = 25), HIV-METH+ (*n* = 25), and HIV+METH+ (*n* = 25). Exclusion criteria included a history of non-HIV-related neurological, medical, or psychiatric disorders that affect brain function (e.g., schizophrenia, traumatic brain injury, epilepsy), learning disabilities, or dementia. Major depressive disorder (MDD) and polysubstance use were minimized but not excluded due to the high prevalence of depression and polysubstance use in Meth users. Table 1 shows demographics, health, and psychiatric characteristics. CSF and plasma viral loads were measured by RT-PCR in a CLIA-certified laboratory. CD4 and CD8 T-lymphocyte counts were measured by flow cytometry for PWH. Nadir CD4 levels were taken from medical records, study-obtained values, or self-report.

**Peripheral leukocytes specimens**—Archived specimens were thawed in fetal bovine serum and washed, and the viability was counted using disposable hemocytometers (Bulldog Bio, Portsmouth, NH, USA). Cell numbers were adjusted for flow cytometry.

**Flow cytometry**—Washed cells had adjusted concentrations of 10^5^/100 μL and were resuspended in HBSS without phenol red, containing 2% fetal bovine serum and 0.2% sodium azide, and stained with predefined concentrations of subset-, function-, and activation-specific antibodies. The surface expression of CCR5, CD38, PDL1, MRP8/14, and RAGE was performed on cells that were incubated from 2 to 24 h with DA or DRD agonists, or on straight cryopreserved human leukocytes from human subjects. The cells were stained with a PerCP-labeled anti-CD11b (clone M1/70, Biolegend, San Diego, CA, USA), APC-labeled antihuman CCR5 (CD195, clone 3A9, BD Pharmingen, San Diego, CA, USA), PE-Cy5-labeled antihuman CD38 (HIT2, Thermo Fisher Scientific), PE-Cy7-labeled antihuman PDL1 (Thermo Fisher Scientific), FITC-labeled antihuman MRP8/14 (Origene, Rockville, MD), and PE-labeled anti-RAGE EPR21171 (Abcam, Wantham, MA, USA). The detection of intracellular p24 was performed with. All the cells were fixed in 4% paraformaldehyde, kept at 4 °C, and protected from light until acquisition. Cell acquisition was performed in a CytoFlex S benchtop platform (Beckman Coulter, Indianapolis, IN, USA) and analyzed using the FlowJo software 10.8.1 for MacOS (FlowJo LLC, Ashland, OR, USA).

**Brain specimens**—Formalin-fixed paraffin-embedded tissue sections of human postmortem basal ganglia specimens were provided by the National NeuroAIDS Tissue Consortium (NNTC) upon request number R593, with Institutional IRB approval (IRB-18-001-MCM). The specimens were selected among all male HIV+ cases (receiving ART) and divided in groups following the same criteria described above, as well as a postmortem interval below 48 h (*n* = 5/group).

**Immunohistochemistry**—Seven µm sections were stained using antibody anti-MRP8/14 (ab17050, mouse antihuman monoclonal 27E10, Abcam, Cambridge, UK), performed as previously described [60]. Colorimetric development was performed with NovaRed chromogen (Vector Laboratories, Burlingame, CA and AbD Serotec, Raleigh, NC, USA, respectively) followed by Gill’s hematoxylin counterstain (Sigma-Aldrich).

**RNAscope in situ hybridization for HIV** (**vRNA**) **detection in brain specimens**—The RNAscope 2.5 HD assay (Advanced Cell Diagnostics, ACD) was performed. Briefly, pretreatment was performed with the RNAscope 2.5 HD Detection Kit (RED) (Cat# 322360), RNAscope 2.5 Pretreat Reagents H202 and Protease Plus (Cat#322330), RNAscope Target Retrieval (Cat#322000), and RNAscope Wash Buffer (Cat#310091), following the manual instructions. The probe set V-HIV1-clade B-C3 (Cat#425531-C3) targeted different segments within the gag–pol region.

**Image processing and quantification**—Images were captured using the light feature of a Zeiss AXIO Observer.Z1 (Carl Zeiss AG, Oberkochen, Germany). Digitalized images in tiff format were opened in Fiji/ImageJ (National Institute of Health, Bethesda, MD, USA) and converted into 8-bit binary masks for quantification and normalization to the total area.

**Statistical analysis**—For the gene expression analysis, scores were calculated as the log2-normalized expression of each gene. Variance due to noise for each score was estimated in a linear mixed model. Pairwise comparisons and the false discovery rate were calculated for each group. Fitted value models were used to determine variable and intercept effects. All biomarker analyses were performed using ANOVA followed by multiple comparisons performed using Tukey’s HD. Predictor screening was used to determine the specific effects of individual variables, named HIV and METH, and their interactions. In human specimens, mixed models were used to identify the interactive effects of the demographics, substance use, detectable cerebrospinal fluid (CSF) viral load, and cognitive global deficit scores (GDSs) on the cell surface markers. All statistical analysis was performed in JMP Pro 15.2.0 software (SAS Institute Inc., Cary, NC, USA).

## 3. Results

### 3.1. HIV Latency in Myeloid Cells Is Partially Reverted by DA Stimulation via DRD1

U1 cells, an established model of HIV latency in the myeloid compartment [51], were used to examine how the neurotransmitter DA affects innate immune phenotypes associated with latency reversal and increased viral replication, mimicking the neuroimmune effects of Meth in the brain. Compared to latency reversal agents such as PEP005, BETi, and PMA, which act on 80 to 100% of all the cells in the culture, DA stimulation at concentrations that mimic levels found in the brain environment during substance abuse [36] consistently promoted replication in 10–15% of the U1 cells as determined by levels of p24 in the supernatant after 24 h (Figure 1A). The same effect was obtained by the selective D1-like receptor (DRD1/5) agonist SKF38393 (SKF), even combined with the DRD2 agonist ropinirole (Rop), which had no effect on p24 levels, or with DRD3 agonist pramipexole (Pra). On the other hand, selective DRD4 stimulation with PD168077 (PD) did not revert latency and prevented reversal by SKF (Figure 2B). Conversely, the stimulation with DA in the presence of a DRD4 antagonist L745 significantly increased p24, while other antagonists had limited or no effect on DA-induced latency reversal (Figure 1C). Together, the results in Figure 1B suggest that D1-like receptors (DRD1/5) and DRD4 have opposing effects on latency. These effects were further characterized.

### 3.2. DA Stimulation via DRD1 Triggers a Novel CD16hi Subset and Novel Transcriptional Signatures

Key surface markers that have been previously described as reporters of activation and inflammation, or in association with changes in replication and myeloid response to DA [44], were measured by flow cytometry with the goal of identifying phenotypic changes that parallel DA-induced increases in p24 levels, which were also present in DRD1/5 selective stimulation. These markers were CD11b, CD14, and CD16 (population and activation markers [61]), as well as CCR5 [44], CD38 [62,63,64], and PDL1 [65] (inflammation, viral replication, activation, and immune senescence). We found that both DA and the D1-like receptor (DRD1/5) agonist increased the levels of CD16 in a subset of cells, which also expressed CCR5. This subset of cells activated by DA comprised 3–5% of the total cells and also coexpressed higher levels of CD38 compared to the controls (Figure 2B,H,N). This observation suggests that exposure of latent cells to DA promotes the development of a subset of latent cells with proinflammatory markers (Figure 2). Importantly, cells with a similar phenotype were also identified in U1 cells stimulated by the D1-like receptor agonist SKF38393 (Figure 2N,O), but not by other DA receptor agonists (Figure 2P,Q,R).

One distinction between the selective stimulation of DA and D1-like receptors (DRD1/5) was related to PDL1 levels, which were predominantly low in CD16high cells following DA, but high in the same subset following SKF38393 stimulation (Figure 2B,C). However, when U1 cells were gated in the CCR5+ population rather than CD16 (Figure 3), we observed that both DA and SKF38393, as well as the stimulation with the DRD2 selective agonist ropinirole, showed increased CD38 (Figure 3B–D) as well as PDL1 expression (Figure 3H–J) in association with CD16hi activation. Cells stimulated with the DRD4 agonist showed a minimal increase in CCR5 or CD38 (Figure 2F,L,R), and also had a low CD38 and PDL1 expression within the CCR5-gated CD16hi subset (Figure 3F,L).

To further characterize changes associated with the onset of a CD16hi novel population, we screened transcriptional signatures triggered by DA via DRD1/5 and in correlation with latency reversal, as well as via DRD4 using gene array. We focused on changes that were common between DA and the DRD1/5 agonist SKF38393, which differed from changes caused by the DRD4 agonist PD168077, which presented significant and opposing effects on p24 levels. Genes that were significantly changed were further tested for their involvement in latency reversal by DA and as signatures of HIV uncontrolled replication in the context of Meth.

To further characterize changes in U1 cells stimulated with DA, or DRD1/5 and DRD4 selective agonists in parallel with changes in p24, we examined global changes in the transcriptome 12 h after stimulation. We found that DA stimulation significantly upregulated eight genes in U1 cells. Of these, the calcium-binding and macrophage-inhibitory-factor-related proteins S100A8 and S100A9, known respectively as myeloid-related protein (MRP) 8 and 14, were significantly increased by both DA and DRD1/5 stimulation (Figure 4A). DA increased MRP8 by 1.67-fold (*p* = 0.007) and DRD1/5 stimulation increased it by 1.89-fold (*p* = 0.005) (Figure 4A). MRP14 was increased by 1.58-fold by both DA and DRD1/5 stimulation (*p* = 0.021 and *p* = 0.029, respectively). Importantly, MRP8 and MRP14 form a heterodimer that was previously described in plasma as a marker of aggravated HIV infection [66], in neurological disorders [67,68], and in proinflammatory responses by endothelial cells [69]. Other genes increased by DA included PDE4B (phosphodiesterase 4B, 1.67-fold), CCL2 (1.52-fold), FTH1 (ferritin heavy polypeptide 1, 1.66-fold), KYNU (kynureninase, 1.68-fold), and HMOX (Heme oxygenase, 2.47-fold) (Figure 4A). DRD1/5 increased RAS dexamethasone-induced protein 1 (RASD1) by 2.8-fold (*p* = 0.00002) as an exclusive signature. DRD4 upregulated 16 exclusive genes: retinoic acid receptor gamma (RARG), NADH dehydrogenase (NDUFAF5, 1.8-fold), midline 2 (MID2, 1.5-fold), ribosomal protein L10 (RPL10, 1.65-fold), RAR-related orphan receptor A (RORA, 1.64-fold), mucin 22 (MUC22, 1.53-fold), MT-RNR2-like 4 (MTRNR2L4, 1.7-fold), translin-associated factor X interacting protein 1 (TSNAXIP1, 2.03-fold), phospholipase A2, group IIE (PLA2G2E, 1.52-fold), ankyrin repeat and SOCS box containing 16 (ASB16, 1.52-fold), disrupted in schizophrenia 1 (DISC1, 1.64-fold), helicase with zinc finger 2, transcriptional coactivator (HELZ2, 1.58-fold), chemokine C-C motif 6 (CCR6, 1.55-fold), DPY19L2, Kruppel-like factor 11 (KLF11, 1.56-fold) and chromosome 14 open reading frame 37 (C14orf37, 1.51-fold). Growth factor receptor-bound protein 10 (GRB10) was increased by both DRD1/5 and DRD4 stimulation (5.02-fold, *p* = 0.03 and 10.03-fold, *p* = 0.000006), but not by DA. The enhancer of mRNA decapping 3 (EDC3) was significantly increased by DA, DRD1/5, as well as DRD4 (2.24-, 5.39-, and 5.26-fold, respectively). Apart from coexpression, pathway-based or functional interactions between these genes that were increased exclusively by DRD4 could not be identified by GeneMania. However, a transcription factor usage prediction using iRegulon suggested that these changes may have occurred with significant contribution of the transcription factors FOXA2 (7.6%), AP4 (7.4%), and GATA1 (7%). Using both the DAVID Bioinformatics Database [44,45,46,47,48,49,50,51,52,53,54,55,56,57,58,59,60,61,62,63,64,65,66,67,68,69,70] and Gene Ontology Resource [71], genes upregulated by DA were annotated to leukocyte migration (*p* = 0.00000002), the positive regulation of inflammation (*p* = 0.0003), inflammatory response (*p* = 0.0008), and the regulation of NFkB transcription factor activity (*p* = 0.00094). Genes upregulated by DRD4 selective stimulation were functionally annotated to transcription initiation from RNA polymerase (*p* = 3.3 × 10^−25^).

We also observed the downregulation of genes by DA, and in particular by DRD1/5 or DRD4 selective stimulation (Figure 4B). DA significantly downregulated a total of nine genes (*p* < 0.05), of which two were exclusive to DA (sulfite oxidase (SUOX, −1.5-fold) and cytochrome P450, family 17, subfamily A, polypeptide 1 (CYP17A1, −1.5-fold)); two were common between DA and DRD1/5 stimulation (olfactory receptor 51I1 (OR51l1, −2.12- and −1.97-fold, respectively) and CUB and sushi multiple domains 2 (CSMD2, −1.5- and −1.58-fold, respectively)); three were common between DA and DRD4 (leucine-rich repeat containing 41 (LRRC41, −3.25-, and −1.79-fold, respectively), solute carrier organic anion transporter family member 1A2 (SLCO1A2, −1.65-, and −1.91-fold, respectively), and solute carrier family 4 member 1 (SLC4A1, −1.65-, and −2.39-fold, respectively)); two were upregulated by all two stimuli (olfactory receptor 52E4 (OR52E4, −1.9-fold, −1.52-fold, and −1.94-fold, respectively) and Ras-protein-specific guanine-nucleotide-releasing factor 1 (RASGRF1, −1.84-, −1.93-, and −2.39-fold, respectively)) (Figure 4B). Interestingly, DRD1/5 stimulation significantly downregulated 92 exclusive genes, which were connected by coexpression, physical and genetic interactions, colocalization, and pathways, as color-coded in the legend (Figure 4C), and were assigned to the Rap1-signaling pathway (*p* = 0.0032). DRD4 downregulated 64 exclusive genes, which also showed interactions (Figure 4D) and were functionally assigned to the regulation of transcription from the RNA Pol II promoter (*p* = 0.002). Moreover, the downregulation of 38 genes was common to both DRD1/5 and DRD4, with identifiable interactions (Figure 4E) and annotated to long-term depression (*p* = 0.0072). Among the genes downregulated by DRD1/5, DRD4, or both, Figure 4B shows a short list of genes with a role in inflammation. See Appendix A for a complete list of all significantly affected genes.

Data integration, analysis, and visualization were performed in the Cytoscape 3.7 software, as previously described [44,45,46,47,48,49,50,51,52,53,54,55,56,57,58,59,60,61,62,63,64,65,66,67,68,69,70]. Pathway-based interactions were sorted using GeneMania [55,56] to identify significantly affected gene networks annotated to biological processes and pathways, confirmed using the DAVID Bioinformatics Database and Gene Ontology Resources, as described [44,45,46,47,48,49,50,51,52,53,54,55,56,57,58,59,60,61,62,63,64,65,66,67,68,69,70]. Figure 5 shows the two most significant networks with overrepresented gene changes as a result of DA, DRD1/5, and DRD4 agonist stimulations combined compared to the vehicle, although with different behaviors. These networks had upregulated (yellow shades) and downregulated genes (blue shades). Genes involved in response to chemical stimulus and the control of transcription through RNA polymerase (*p* < 0.0004) (Figure 5A–C) and inflammation and chemokine receptors (*p* < 0.00001) (Figure 5D–F) exhibited similarities between DA and DRD1/5 stimulation, but also differences. Other identified pathways included cell motility (*p* < 0.0004), as well as neurotransmission and cardiac function (*p* < 0.0005), with overlapping representation with the networks in Figure 5. The pathway annotated to the response to the stimulus and the regulation of transcription (Figure 5A–C) showed differences in the expression of specific genes that paralleled the reversed latency by DA and DRD1/5 stimulation, and replicated the visualization in Figure 4, with increased levels of S100A8 and S100A9, or MRP8 and MRP14, by DA and DRD1/5 (Figure 5A,B). The visualization of gene interactions in Figure 5A–C suggests that the transcriptional levels of MRP8 and MRP14 were inversely correlated with the transcription of the retinoic acid receptor gamma (RARG) levels. Regarding the pathway annotated to inflammation and to chemokine receptor (Figure 5D–F), the expression patterns were mixed.

### 3.3. MRP8 and MRP14 Are DA Signatures That Reverse HIV Latency via RAGE

Given that both DA and the DRD1/5 agonist increased the transcription of MRP8 and MRP14, we examined whether the heterodimer is induced at the protein level in U1 cells following DA exposure (Figure 6) [72,73]. We found that DA both at 1 μM and at 10 μM was able to significantly increase the secretion of MRP8/14 in the U1 supernatant, detectable at 12 h after stimulation. Interestingly, the DRD1/5 agonist at the lowest concentration of 0.1 μM, and the DRD4 agonist at the highest 1 μM concentration, were able to trigger the release of MRP8/14 (Figure 6A). Other selective agonists did not have an effect (not shown). Similarly, the concentration of MRP8/14 in the cytoplasm was increased by DA via both receptors (Figure 6B). The increase in the MRP8/14 complex was also observed on the cell surface following DA stimulation (both 1 and 10 μM), as well as with both DRD1/5 and DRD4 selective agonists (Figure 6B,C), but not with other DRD agonists (not shown). Interestingly, the stimulation via DRD4 significantly decreased the surface expression of the MRP8/14 receptor RAGE [74] (Figure 6F). Neither DA nor the DRD1/5 agonist SKF389393 affected the expression of the RAGE on the cell surface (Figure 6F). To test the significance of these findings for latency, we examined whether MRP8/14 was able to increase the expression p24 in U1 cells, mimicking the effects of DA and SKF38393 on p24 (Figure 6G). For that, U1 cells were exposed to different doses of the recombinant MRP8/14. We also tested the effect of MRP8/14 via its two described receptors, the receptor for advanced glycation end-products or RAGE [74] and toll-like receptor 4 or TLR4 [75,76]. Indeed, MRP8/14 was able to increase the levels of p24 detectable at 24 h in a dose-dependent manner (Figure 6G). The inhibition of RAGE by a selective antagonist FPS-ZM1 [77] also prevented the effect of MRP8/14 on p24 in a dose-dependent manner (Figure 6G) without affecting cell viability (not shown). The selective TLR4 inhibitor TAK242, on the other hand, did not inhibit the MRP8/14-induced increase in p24, unless at a higher dose of 10 μM, when it also decreased cell viability from 98% (±7) to 56% (±10) (*p* = 0.003). This suggested that MRP8/14 are the transcriptional signatures induced by DA that influence latency, reversing it via binding to RAGE. Interestingly, although both DRD1/5 and DRD4 selective agonists increased secreted, cytosol, and surface MRP8/14, DRD4 selective stimulation caused a drastic decrease in the RAGE expression (Figure 6F), indicating its ability to prevent activation and latency reversal via RAGE. This observation could provide a mechanistic link to explain the relative differences in the latency response to DA and why DRD4 selective stimulation may prevent latency reversal, as seen in Figure 1.

### 3.4. DA Signatures Expressed on Peripheral Innate Immune Cell Surface Are Biomarkers of HIV Infection and Detectable CSF Viral Load in the Context of Meth Use

We examined the translational value of our in vitro findings by examining whether the observed signatures triggered by DA could be identified in human subjects and whether they can distinguish HIV+ individuals that are Meth users compared to non-Meth users (Figure 7). Moreover, given the involvement of these biomarkers in latency reversal, we compared individuals with detectable or undetectable viral load in the cerebrospinal fluid (CSF) at the time-point of specimen collection. For that, specimens from 102 individuals from the Translational Methamphetamine Research Center (TMARC) cohorts (UCSD) were distributed in 4 groups, as shown in Table 1, with HIV−METH− (*n* = 27), HIV+ METH− (*n* = 25), HIV−METH+ (*n* = 25), and HIV+ METH+ (*n* = 25) subjects, homogeneous for sex (all males due to the characteristics of the cohort and sample size), age, and education to minimize confounders. Table 1 shows the demographics and other characteristics of the cohort, including the percentage of individuals with detectable viral load in plasma and CSF. We tested the expression of CD38, CCR5, PDL1, RAGE, and MRP8/14 by flow cytometry on the surface of peripheral blood leukocytes.

The value of these markers varied to indicate effects of HIV, Meth, or their interaction (Figure 7), as well as to distinguish the group with the detectable CSF viral load, particularly within the HIV+METH+ group (Figure 8). For instance, CD38 expression was overall higher in subjects of the HIV + METH+ group with a detectable CSF viral load compared to an undetectable viral load within the same group (*p* = 0.043) (Figure 8A), although this marker was not overall affected by HIV or Meth (Figure 7A). While CCR5 did not distinguish any variables (Figure 7B and Figure 8B), PDL1 expression was significantly decreased by HIV infection regardless of Meth status (Figure 7C). MRP8/14 was lower in CD16hi cells from HIV− Meth users (Figure 7D). In HIV+ subjects, an interactive effect of HIV, Meth, and the detectable CSF viral load was indicated, with a trend to a higher expression of MRP8/14 in Meth users with the detectable viral load. A similar trend was observed with the expression of RAGE (Figure 8E).

We also tested whether the expression of combined markers could detect subject groups more efficiently than single markers (Figure 9). We found that HIV+ METH+ subjects with undetectable CSF viral loads had significantly lower CCR5 expression in the CD11b-gated CD16hi CD38hi subset compared to HIV+ METH+ subjects with a detectable viral load (*p* = 0.0021), as well as to HIV+ METH− subjects with an undetectable viral load (Figure 9A). We also found that the HIV+ METH+ group with a detectable CSF viral load had a significantly lower PDL1 expression in the same cell subset compared to subjects with an undetectable viral load in the CSF (*p* = 0.003) or with METH− groups (*p* = 0.03 and *p* < 0.0001, detectable and undetectable CSF viral loads, respectively) (Figure 9B).

### 3.5. MRP8/14 and HIV RNA Are Increased in HIV-Infected Postmortem Brain Specimens in Association with Meth Use

We also examined whether MRP8/14+ cells could be identified in the brains of HIV+ Meth users, using postmortem brain specimens of basal ganglia obtained from the National NeuroAIDS Tissue Consortium (NNTC). For that, HIV−METH− controls (*n* = 5), and HIV+ METH− (*n* = 5), HIV-METH+ (*n* = 5), and HIV+ METH+ (*n* = 5) specimens (five sections per subject), selected to display minimal confounders and no reported polysubstance use, were stained using immunohistochemistry to detect MRP8/14+ cells. Representative sections from each group (Figure 10A), and the assessment of MRP8/14+ cell numbers per area (Figure 10B), indicated that both HIV and Meth alone significantly increased the MRP8/14 cell numbers by 2- and 3-fold, respectively, compared to the controls, while HIV+ METH+ had a significant > 4-fold increase in this subset. This suggests that Meth use is a factor in further increasing MRP8/14+ cells in HIV+ brains. Moreover, by RNAscope, we observed an increase in the detectable viral RNA, in correlation with Meth use, in the basal ganglia (Figure 10C). This indicates that MRP8/14 may be a brain biomarker linked to viral replication in the context of Meth use in humans.

## 4. Discussion

This is study is the first report of a link between DA-induced signatures in innate immune cells and phenotypes of HIV latency reversal as biomarkers of CNS infection in the context of Meth-use disorder.

The in vitro latent promonocyte cell-line model has shown that DA has a moderate but significant effect on reversing HIV proviral latency, while promoting the development of a subset of cells expressing proinflammatory markers. The cell subset that appears in correlation with a higher p24 is CCR5high, CD38high, and PDL1low within the CD16high-activated subset. We have previously identified cells with similar characteristics in the brains of SIV-infected rhesus macaques that were chronically treated with Meth in correlation with a higher brain viral load [4]. The percentage of cells that upregulate these markers matches the calculated percentage of virus-replicating cells following DA- or DRD1/5-selective stimulation. However, it is not clear from these experiments whether they correspond to the same cells. Yet, these are molecules with described consequences to inflammation and in HIV infection, as well as in neurological disorders. For instance, the increase in CCR5, which increased the expression in peripheral cells, has been described as a biomarker of Meth [4,44,78,79].

CD38 is a multifunctional enzyme expressed on the cell surface that plays a major role in the hydrolysis of NAD, and consequently regulates Sirt1 enzymatic activity [21,80,81]. We have shown evidence that latently infected myeloid cells exposed to DA experience the onset of a novel subset expressing high levels of CD38 and CCR5 in correlation with latency reversal. PDL1, on the other hand, indicates cell senescence [82]. A subset with low levels of PDL1 may have a long-living phenotype, indicating reservoir retention. We also show that a subset of CCR5high, CD38high, and PDL1low within the CD16high subset is increased in the peripheral blood from PWH that are Meth users.

Transcriptionally, the stimulations with DA, DRD1/5, or DRD4 agonists predominantly caused the downregulation of genes rather than upregulation within the latently infected myeloid cells. Interestingly, we have previously observed that Meth administration to mice triggers strong suppression of gene expression in the brain, although it seems to be counteracted by the expression of Tat in the astrocytic compartment [83]. Here, the in vitro data suggest that, in the isolated latently infected compartment, DA as a major neurotransmitter in the brains of Meth users may contribute to the retainment of low gene transcription profiles while stimulating pathways that favor viral replication secondarily or as a result of inflammation. DA increased few markers involved in stimulus response and transcriptional regulation, which also have described effects on inflammatory outcomes. Two of these transcriptional signatures were associated with latency reversal via DRD1/5 in our model, MRP8 and MRP14, which form a heterodimer at the protein level [73], also known as calprotectin. Interestingly, the expression of MRP14 and the heterodimer have been previously shown to serve as a signature of a macrophage subset that is not infected but is enriched in the perivascular domain in HIV and SIV encephalitis [84]. We have validated the role of DA in the upregulation of the MRP8/14 heterodimer, both secreted and on the cell surface, via specific receptors, indicating that this may be a marker of CNS disorder in drug users, as well as a factor triggering replication in latent cells via binding to its receptor, RAGE, on infected cells.

Overall, few reports link MRP8 and/or MRP14, alone or as a heterodimer, with aggravated HIV pathogenesis [66,67,68,69,70,71,72,73,74,75,76,77,78,79,80,81,82,83,84,85] associated to CD4 T cells [86] or with early monocyte migration [84]. A recent report has linked MRP8 (S100A8) expression to active HIV transcription and metabolism in cultured urethral macrophages, along with other inflammatory markers such as IL1R and MMP7 [87]. One report has found the direct effects of MRP8 in HIV replication [88]. One report links the expression of the heterodimer in mucosa with viral shedding independent from the plasma viral load [89], suggesting localized effects. On the other hand, no effects of the MRP8/14 complex on HIV infection, or the suppression of p24 levels by subunits, have been reported in models where infected cells are differentiated and activated at baseline [90]. These reports were observed using approaches that differ significantly from the approach in our study, as here we attempted to address the effects on infected cells exhibiting a latent phenotype, which is relevant in the post-ART era. It also suggests that MRP8 and MRP14 may serve as latency reversal cytokines, but may not enhance ongoing replication, or may even be beneficial in the context of ART. MRP8 and MRP14 have been identified in other infections, such as SARS-CoV-2, in association with high inflammatory levels [90]. In conditions of coinfection or comorbidities, these cytokines could act differently. Moreover, it has been reported that the MRP8/14 complex is increased in the cerebrospinal fluid of HIV+ subjects exhibiting coinfections [91]. It has never been examined in the context of SUD. It remains to be shown how DA affects actively replicating cells. In addition, the effects of MRP8/14 may differ by the activation state, in part due to binding to different receptors.

The receptor for MRP8/14-mediated effects on HIV replication in latent cells was identified as RAGE [92,93], a pattern-recognition receptor that decreases in expression and function during aging, resulting in higher oxidative stress and with implications for vascular calcification, diabetes, dementia, and other aging consequences [94,95,96]. Interestingly, RAGE has been described as a target for Alzheimer’s disease improvement and the prevention of amyloid deposition in the CNS [97,98,99,100,101,102], while increasing CCR5 expression and promoting leukocyte transmigration across the blood brain barrier [103]. Studies in TLR4 and RAGE knockout models have suggested a major role of TLR4 as a receptor to MRP8/14 on myeloid cells [76]. However, we did not find compelling evidence for TLR4 in latency reversal by MRP8/14 in our model. In HIV infection, low levels of soluble RAGE have been found in infected subjects with subclinical carotid atherosclerosis under ART [104], but also complex protective effects against cys-infection from myeloid dendritic cells due to CCR5 repression, in spite of the induction of proinflammatory cytokines [105]. In our model, where the ligands to RAGE are upregulated by DA and RAGE is involved in latency reversal, higher CCR5 expression is one of the detectable signatures in a subset of the cells and during chronic Meth use in vivo [4,44,78]. Perturbations to RAGE ligands, such as monoamines, have been involved in CNS conditions, including aging and neurodegeneration, and the impairment of antioxidant mechanisms [106,107,108,109,110]. An interesting study in flies has shown that feeding the animals with cocaine or Meth, as well as exposing them to DA, increases the levels of advanced glycation end-products and RAGE ligands. Moreover, mutant flies lacking the dopaminergic transporter or DRD1 have shown the accumulation of these ligands [111], further suggesting the role of RAGE in DA-induced perturbations. Here, we show that DA exposure promotes an increase in a myeloid-related complex, MRP8/14, that serves as a RAGE ligand in latently infected human myeloid cells. The results suggest that these molecules have a biomarker value and play a role in HIV replication linked to inflammation. We have also shown that cells expressing these molecules are enriched in the brains of HIV+ individuals, and even further in Meth users, suggesting a mechanism for a DA-mediated increase in the viral load in the brains of HIV+ individuals with substance-use disorders. Relevant to the brain, MRP8 has also been previously implicated in neurological outcomes, such as depression [68], inflammation in the CNS [86,112,113,114,115], and elsewhere [72,116,117,118,119], as a mediator and biomarker.

The differential effects of DRDs on the onset of subset populations in correlation with levels of p24 were intriguing, particularly the effects of DRD4 versus DRD1/5. We show evidence that the RAGE ligand MRP8/14 heterodimer is increased in vitro by DA and in vivo by Meth, and that RAGE surface expression on latent myeloid cells is decreased by DRD4 by means that may actively maintain latency. RAGE is also expressed at lower levels on monocytes from HIV+ Meth users with an undetectable viral load. Interestingly, DRD4 is a gene with an unusually high degree of polymorphism, with some variants increasing the risk of substance-use disorders [120,121,122,123]. The implications of this polymorphism for immune responses in the context of substance-use or neurological disorders have not been examined.

Although it is known that immune functions interfere with neuronal functions and vice versa, specific DRD subtypes, immune activation, and neurological diseases have been rarely linked. It has been shown that DA signals via DRD4 to promote Th2 responses, with implications for allergic reactions [124]. In microglia, a decrease in the inflammatory activation has been described in hyperdopaminergic conditions, attributed to a decrease in nitric oxide [125] and phagocytic capacity [126], potentially mediated by DRD1 and DRD2 [127]. It has also been suggested that DRD1 signaling in microglia induces the degradation of the NLRP3 inflammasome in a cAMP-dependent fashion in vivo [128]. Here, we show that the downregulation of genes in latent innate immune cells is more significant than the upregulation, both via DRD1/5 and via DRD4, with distinct and common signatures. Yet, we also show that, among a few upregulated genes, proinflammatory ones are expressed. Similarly, the administration of Meth in mouse models of neuroHIV has led to the observation of the drastic suppression of gene expression, counteracted by the expression of the HIV accessory protein Tat, and with a focal increase in specific inflammatory pathways [83].

The combined expression of the markers increased by DA via DRD1/5 in a subset of cells, CCR5, CD38, and PDL1, along with MRP8/14 and RAGE, have indicated the strong potential to distinguish PWH that are Meth users with a detectable and undetectable CSF viral load. We did not find a correlation with the plasma viral load, further suggesting the CNS-localized effect of the proposed mechanism of latency reversal involving the upregulation of MRP8/14 and its binding to RAGE, triggered by high levels of DA in the context of Meth use. The analysis of males, which was due to cohort characteristics and to increase the power, with an *n* = 25, and the lack of a precise time definition for lifetime criteria, are limitations. Yet, the study addresses the value of biomarkers considering demographic trends and a gender disparity in the population of individuals with SUD and HIV [129].

Interactions between HIV and neurotransmitters induced in the brain by addictive stimulant substances must be better understood for addressing consequences, including the aggravated damage to areas bearing dopaminergic projections, such as the basal ganglia [130,131], and the neuropathologic effects of glial cells and inflammation [132]. Specific signatures of virus-infected cells in the context of a hyperdopaminergic microenvironment may explain the variabilities of inflammatory outcomes in the context of substance use and provide mechanisms to explain the changes in replication, as well as aggravation, that are found in Meth users.

## Figures and Tables

**Figure 1 viruses-15-01363-f001:**
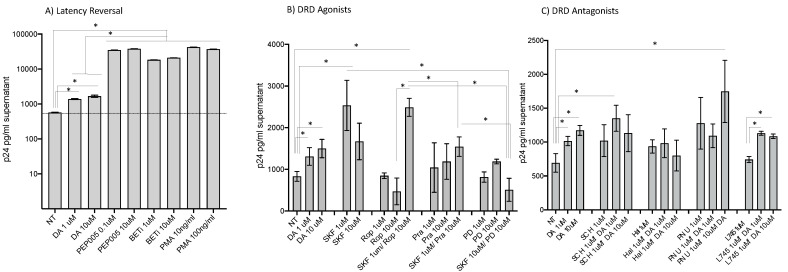
**Effects of DA and DRDs on p24 levels.** HIV p24 levels were measured by ELISA in U1 cell culture supernatants 24 h following the stimulation with (**A**) DA and classic latency reversal agents PEP005 (0.1 and 10 μM), BETi (1 and 10 μM), and PMA (10 and 100 ng/mL). (**B**) Selective DRD agonists SKF38393 (SKF, DRD1/5; 1 and 10 μM), ropinirole (Rop, DRD2; 1 and 10 μM), pramipexole (Pra, DRD3; 1 and 10 μM), and PD168077 (PD, DRD4; 1 and 10 μM). (**C**) Selective DRD antagonists SCH23390 (SCH, DRD1/5), haloperidol hydrochloride (Hal, DRD2), PNU177864 (PNU, DRD3), and L745870 hydrochloride (L745, DRD4) were added at 1 μM to cultures, alone or combined with 1 and 10 μM DA. Representative experiments out of 3 independent assays performed in quadruplicate show the mean ± SD. NT = nontreated vehicle control. * *p* < 0.05 in ANOVA followed by Tukey’s HSD.

**Figure 2 viruses-15-01363-f002:**
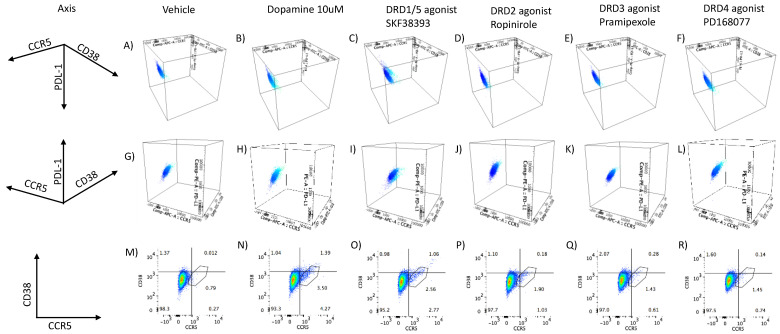
**Surface markers of population and activation status in latent U1 cells treated with DA and DRD agonists.** The expression of population and activation markers was measured on the U1 cell surface by flow cytometry upon exposure to DA and selective receptor agonists. U1 cells were either (**A**,**G**,**M**) in control conditions (vehicle/nontreated), (**B**,**H**,**N**) treated with 10 μM of DA, or treated 1 μM of DRD selective agonists (**C**,**I**,**O**) SKF38393 (DRD1/5), (**D**,**J**,**P**) ropinirole (DRD2), (**E**,**K**,**Q**) pramipexole (DRD3), or (**F**,**L**,**R**) PD168077 (DRD4). The plots are representative of 3 independent experiments. (**A**–**L**) The CD14+ CD16low subset in shown in dark blue and the CD14+ CD16 high-activated subset, shown in light blue, displaying a three-dimensional distribution of CCR5 (*x*-axis, PDL1 (*y*-axis), and CD38 (*z*-axis). (**M**–**R**) Flat scatter plots of surface CCR5 (*x*-axis) and CD38 (*y*-axis) in ungated live cells. A gate on the CCR5+ CD38+ was used to estimate the percentage of the total cells expressing these markers.

**Figure 3 viruses-15-01363-f003:**
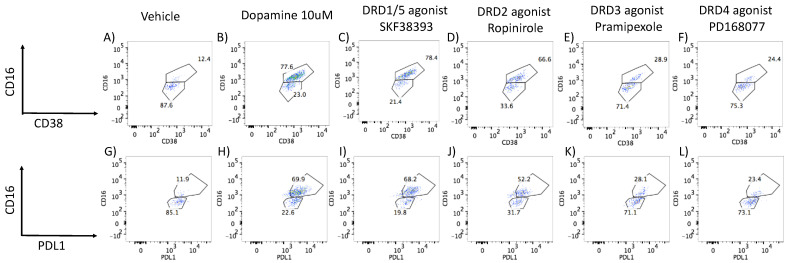
**Surface markers of inflammation in gated CCR5+ U1 cells treated with DA and DRD agonists.** The expression of population and activation markers was measured by flow cytometry on U1 cells upon exposure to DA and selective receptor agonists. The markers examined in combination were (**A**–**F**) CD16 vs. CD38 or (**G**–**L**) CD16 vs. PDL1. U1 cells were either (**A**,**G**) in control conditions (vehicle/nontreated), (**B**,**H**) treated with 10 μM of DA, or treated 1 μM of DRD selective agonists (**C**,**I**) SKF38393 (DRD1/5), (**D**,**J**) ropinirole (DRD2), (**E**,**K**) pramipexole (DRD3), or (**F**,**L**) PD168077 (DRD4). The plots are representative of 3 independent experiments.

**Figure 4 viruses-15-01363-f004:**
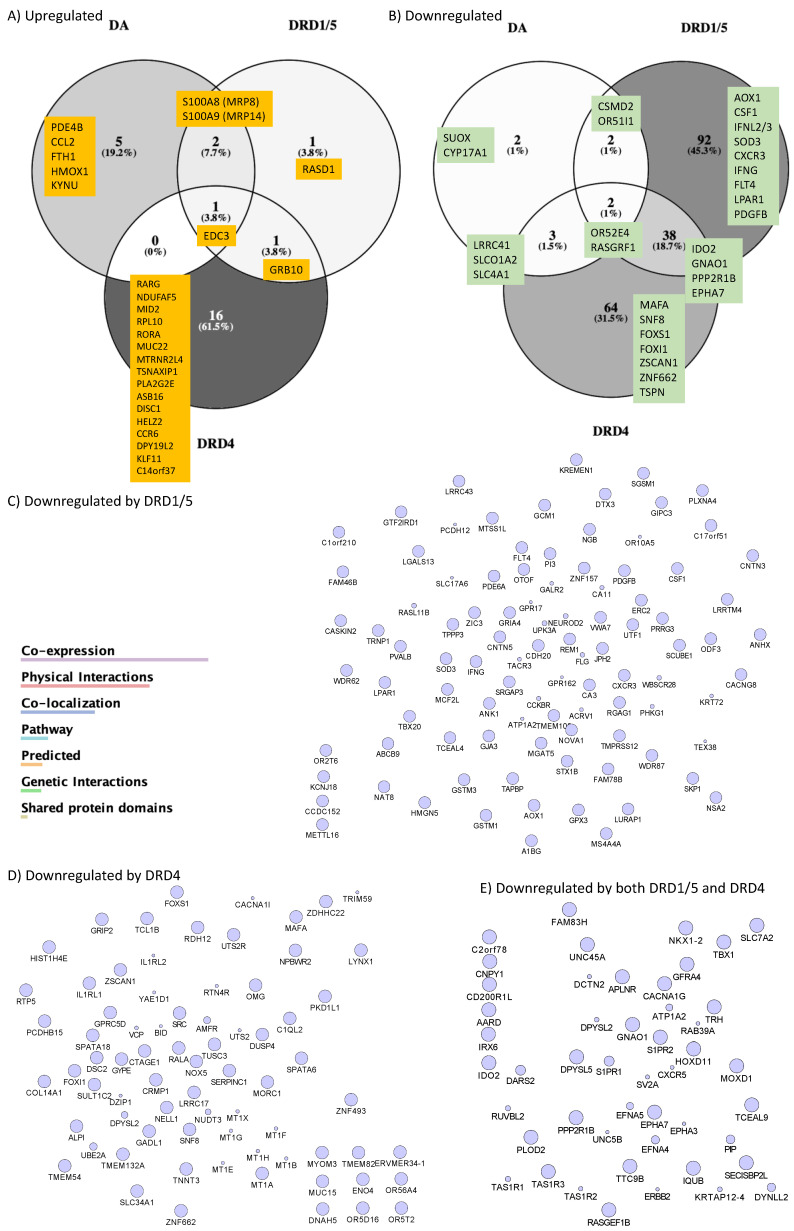
**Venn diagrams showing gene signatures identified in response to DA and DA receptor agonists and in correlation with replication** (**DA and DRD1/5 agonist SKF38393**) **or latency** (**DRD4 agonist PD168077**)**.** Following group comparisons for each individual gene, log10 fold change between DA and vehicle (DA), SKF38393 and vehicle (DRD1/5), or PD168077 and vehicle (DRD4). Significantly changed genes (FDR *p* < 0.05) were selected and a cut off of 1.5-fold was used to identify genes that were (**A**) upregulated or (**B**) downregulated by stimuli compared to controls. Venn diagrams indicate the genes fitting these criteria in each group and genes that overlapped between stimuli. When the number of genes in a given comparison was too large to be listed, the number of genes was scripted and the 10 most downregulated were listed. Systems biology and visualization strategies were used to determine the relationship and interactions between the genes downregulated by DRD1/5 and DRD4, as well as by both. The legend shows connectors’ colors indicating the different levels of interactions between genes. (**C**) Interactions between the genes downregulated exclusively by DRD1/5 selective stimulation. (**D**) Interactions between genes exclusively downregulated by DRD4. (**E**) Interactions between genes commonly downregulated by DRD1/5 and DRD4 selective stimulation. Genes that are changed but do not have reported interactions and do not show connectors. A complete list of differences can be found in Appendix A.

**Figure 5 viruses-15-01363-f005:**
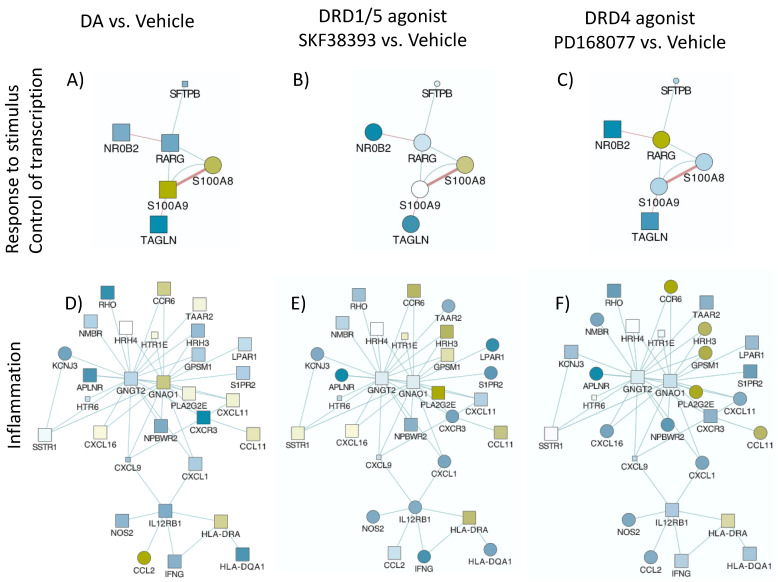
**Pathways significantly affected by DA and by selective DRD1/5 and DRD4 agonists.** U1 cells stimulated with (**A**,**D**) 10 μM DA, (**B**,**E**) D1-like DRD (DRD1/5) agonist SKF389393, or (**C**,**F**) with the DRD4 agonist PD168077 were compared to vehicle-treated, nonstimulated cells. Transcriptional changes were examined by the Agilent Whole Human genome platform. Normalized values were compared using log fold changes between conditions, and then visualized using Genemania in the Cytoscape 3.3 interface, with pathway-based (blue lines) and shared protein-domain-based interactions (red lines). Blue shapes represent downregulation and yellow shapes represent upregulation. Circles are *p* < 0.05 compared to the vehicle. Two main gene networks were identified and annotated to (**A**–**C**), response to the stimulus and control of transcription, and (**D**–**F**) inflammation.

**Figure 6 viruses-15-01363-f006:**
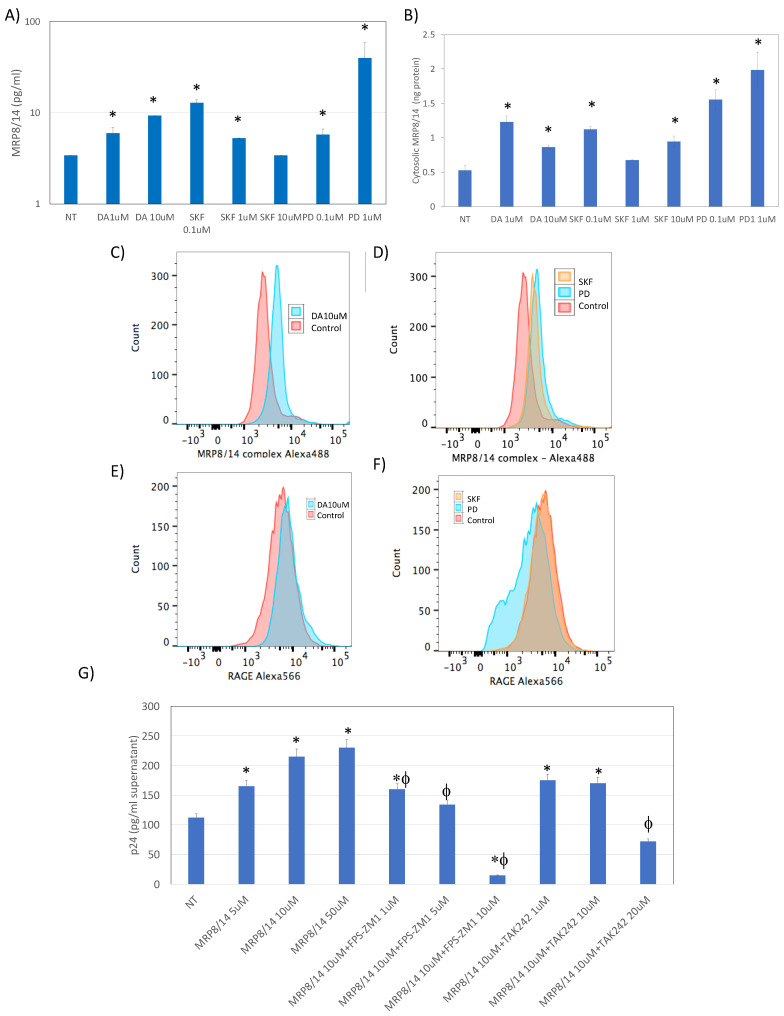
**MRP8/14 is upregulated by DA and reverses latency via RAGE.** (**A**) Levels of MRP8/14 in culture supernatants measured by bead-based immunoassay 24 h after stimulation with DA, with the DRD1/5 agonist SKF38393 (SKF) or the DRD4 agonist PD168077 (PD) at the indicated concentrations, normalized by volume, and calculated using the standard curve. (**B**) Concentration of MRP8/14 in the cytosolic fraction extracted 24 h after stimulation, normalized to total protein. (**C**–**F**) The surface levels of the DA-induced biomarkers were detectable by flow cytometry. Representative histograms indicate the geometric mean fluorescence (GeoMean) of (**C**,**D**) MRP8/14 and (**E**,**F**) RAGE in cells stimulated with (**C**,**E**) 10 μM DA (**D**,**F**) with the DRD1/5 agonist SKF389393 or with the DRD4 agonist PD168077 compared to the controls, as indicated in the legend. (**G**) HIV p24 levels in the culture supernatant measured by ELISA 24 h after stimulation with recombinant MRP8/14 at indicated doses, in the presence or absence of FPS-ZM1 (RAGE antagonist) or TAK242 (TLR4 antagonist). Tukey’s HD multiple comparisons; * *p* < 0.05 compared to nontreated (NT) control; ϕ *p* < 0.05 compared to 1 μM MRP8/14.

**Figure 7 viruses-15-01363-f007:**
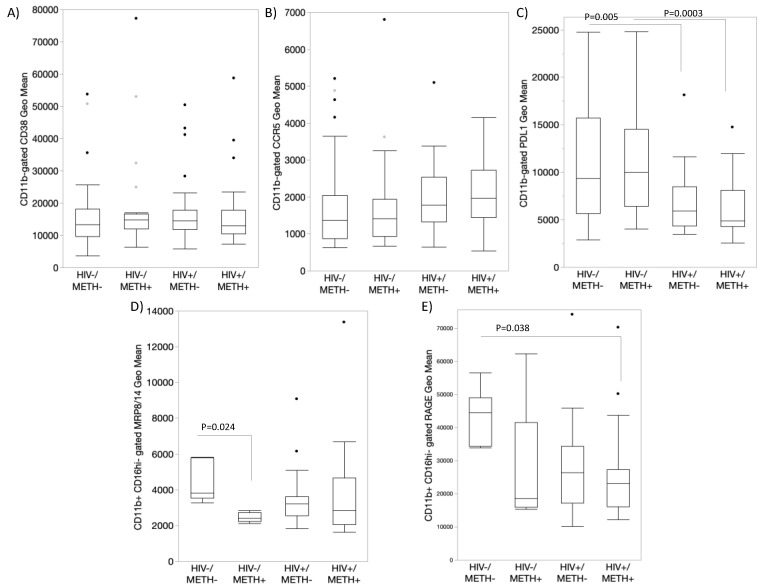
**Characterization of peripheral monocytes from HIV+ Meth users using surface expression of DA signature proteins.** Using flow cytometry, we measured the expression of DA signatures and inflammatory markers in peripheral blood monocytes from HIV+ and HIV− subjects that were Meth users (METH+) or not (METH−). The monocytes were gated using CD11b, CD14, and CD16 antibodies, and the geometric mean fluorescence of (**A**) CD38, (**B**) CCR5, (**C**) PDL1, (**D**) MRP8/14, and (**E**) RAGE were calculated in the FlowJo v10.8.1. Multiple comparisons were performed using Tukey’s HD, with indicated *p* values. Dots indicate outliers. Mixed models indicated effects of HIV, METH, and CSF viral load interactions.

**Figure 8 viruses-15-01363-f008:**
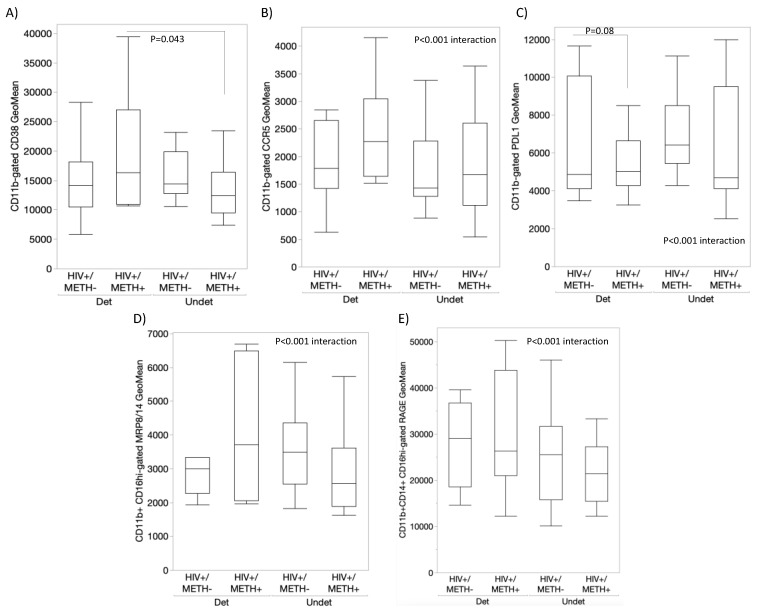
**Surface expression of DA signatures on monocytes from HIV+ Meth and CSF viral load.** Using flow cytometry, we measured the expression of DA signatures and inflammatory markers in peripheral blood monocytes from HIV+ subjects that were Meth users (METH+) or not (METH−), further divided into groups with a detectable or undetectable CSF viral load. The monocytes were gated using CD11b, CD14, and CD16 antibodies, and the geometric mean fluorescence of (**A**) CD38, (**B**) CCR5, (**C**) PDL1, (**D**) MRP8/14, and (**E**) RAGE were calculated in the FlowJo software v10.8.1. Multiple comparisons were performed using Tukey’s HD, with indicated *p* values. Mixed models indicated effects of HIV, METH, and CSF viral load interactions.

**Figure 9 viruses-15-01363-f009:**
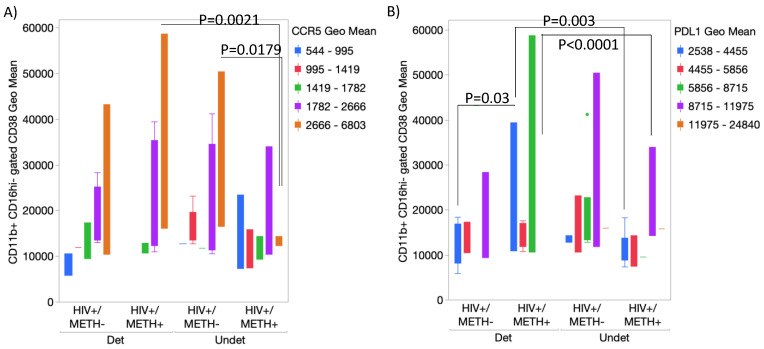
**Combined expression of surface markers on peripheral monocytes.** Using flow cytometry, CD11b+ CD16hi-activated cells were gated and further analyzed for their expression levels of CD38 combined with (**A**) CCR5 and (**B**) PDL1. Bar colors represent the expression level ranges, as indicated in the legends. Multiple comparisons were performed using Tukey’s HD, with indicated *p* values. Mixed models indicated effects of HIV, METH, and CSF viral load interactions.

**Figure 10 viruses-15-01363-f010:**
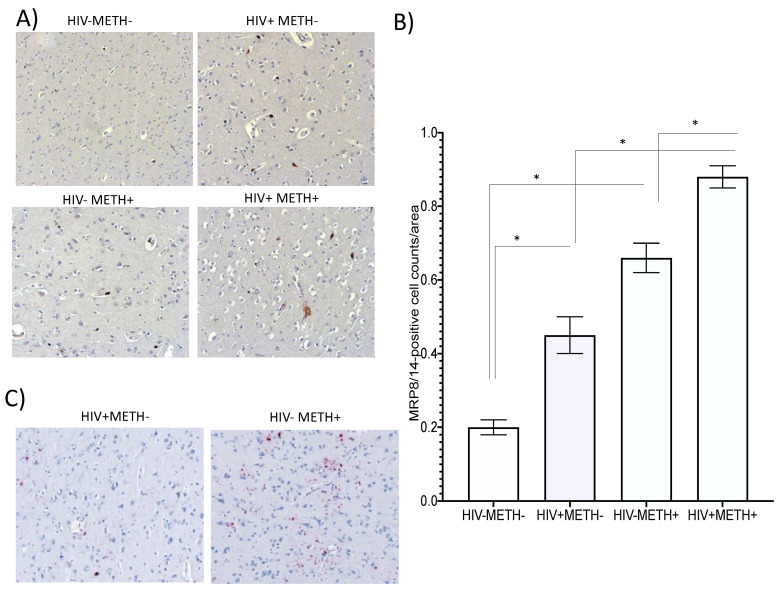
**Detection of MRP8/14 in human brain specimens of HIV+ Meth users.** Postmortem formalin-fixed paraffin-embedded human brain specimen sections were obtained from the NNTC. Immunohistochemistry was used to detect MRP8/14-positive cells in the basal ganglia. (**A**) Representative sections from HIV-METH-, HIV+ METH-, HIV-METH+, and HIV+ METH+ brains. (**B**) The average number of MRP8/14-positive events was counted in 25 fields per section, in 5 sections per subject, and in 5 subjects per group, using 40× magnification. (**C**) RNAscope in situ hybridization was used to detect HIV RNA in the same sections. The pictures show a representative section from HIV+ METH− and HIV+ METH+ groups. * *p* < 0.05 in indicated comparisons.

**Table 1 viruses-15-01363-t001:** Cohort demographic, health, and psychiatric characteristics.

	HIV-/METH−		HIV+/METH−		HIV−/METH+		HIV+/METH+		ANOVA *p* Values
N	27		25		25		25		
	Mean	STD	Mean	STD	Mean	STD	Mean	STD	
**Age**	37.68	9.17	38.8	6.94	36.16	9.26	36.32	6.44	0.4980
**Education**	13.68	2.37	13.24	2.61	12.7	2.45	12.08	2.81	0.0882
**Global T score**	49.70	6.64	46.19	5.66	45.94	5.7	46.17	6.77	0.0524
**CD4 Nadir**	815	205	287	219	918	184	311	0.23	<0.0001
**CD4/CD8 Ratio**	1.84	1.29	0.53	0.37	2.48	1.28	0.56	0.26	<0.0001
**Duration of infection (yrs)**	NA	NA	7.26	5.73	NA	NA	6.18	6.98	0.4431
**% Black**	7		3		3		0		
**% Hispanic**	5		6		2		5		
**% Asian**	1		1		1		2		
**% White**	14		15		19		18		
**Detectable Plasma VL (% of total)**	NA		60		NA		56		0.0774 *
**Detectable CSF VL (% of total)**	NA		48		NA		58		0.0293 *
**LT Alcohol dep (% of total)**	60		45		68		52		0.0407
**LT Cocaine dep (% of total)**	20		32		36		24		
**LT Opioid dep (% of total)**	4		4.5		24		12		<0.0001
**LT MDD (% of total)**	16		72		48		52		

VL = viral load; LT = lifetime, reported three or more DSM-IV criteria in their life; dep = dependence; MDD = major depressive disorder; NA = not applicable. * The HIV+ subjects did not differ significantly in the number of years receiving ART (*p* = 0.137).

## Data Availability

All raw data has been made available in GEO number GSE221531. Additional data can be made available upon request.

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
