# Peer review of "MRP8/14 Is a Molecular Signature Triggered by Dopamine in HIV Latent Myeloid Targets That Increases HIV Transcription and Distinguishes HIV+ Methamphetamine Users with Detectable CSF Viral Load and Brain Pathology"

_viruses, 2023, doi:10.3390/v15061363_

Round 1

Reviewer 1 Report (New Reviewer)

I feel the authors have improved their manuscript and satisfactorily addressed my comments. I do appreciate the high-quality figures which the pdf conversion process did not seem to alter this time.

I just have two petty comments which won’t preclude me from recommending acceptance to the Editor:

- Authors could consider changing “peptide” for “protein” in the following sentences: “For instance, HIV peptides such as Tat” and “is altered by the virus or its peptides”

- Would NA in Table 1 be better defined as not applicable? I suggest the authors to consider this change as well.

Author Response

Thank you so much for the comments.

We appreciate the time taken to review the manuscript.

We have modified the requested places.

Reviewer 2 Report (New Reviewer)

This is a study investigating the effects of dopamine in HIV myeloid targets and HIV transcription. The strengths of this study are: the inclusion of both in vitro and in vivo experimentation. Limitation: relatively small sample size. My comments are as follows: 

1. Clarify in the Abstract that the brain studies were conducted as 'post-mortem analyses'; That is, "... we tested its expression in HIV+ Meth users' brains using post-mortem specimens and peripheral cells"

2. Please clarify 'Lifetime cocaine dependence' in the Table. Is there a specific duration of meth dependence in the cohorts assessed?

3. Include Limitation: relatively small sample size

Author Response

We have highlighted the parts where suggestions were included.

We appreciate your comments.

This manuscript is a resubmission of an earlier submission. The following is a list of the peer review reports and author responses from that submission.

Round 1

Reviewer 1 Report

The manuscript by Basova et al. describes the analysis of the transcriptome of latent cells stimulated with dopamine (DA), via DRD1 and DRD4, leading to the identification of S100A8 and S100A9 as essential genes correlated with early elevation of p24 expression following DA. These genes are translated into MRP (8 and 14), forming a complex MRP8/14 that was upregulated by DA on the innate immune cell surface. They have shown that in human brains from PWH that were Meth users, MRP8/14 was upregulated, suggesting MRP8/14 as a signature of substance use with the potential of aggravating neuroinflammation and viral replication in the brains of PWH on Meth. Although the study is a great study in the field of HIV infection and substance use disorder (SUD) comorbidity, yet authors should provide some points that should be addressed while correcting the manuscript:

1.          It is unclear if U1 (promonocytic) cells were differentiated into macrophages or used as promonocytes. If they were not differentiated, then authors need to explain how they can claim it as a macrophage-driven study.

2.          In this study, the authors used only male participants for the different experimental groups. Although SUD is more common in males than females, authors need to explain the reasoning behind choosing only males and how this study can be relevant to the general population.

3.          While grouping in method section lane 160, the authors incorrectly named the group HIV+CAN+.

4.          In figure 3 it seems the authors have missed putting the figures for M, N O P, and R. Also, Fig 6C looks distorted and difficult to draw any conclusion from.

5.          I also feel the abstract can be improved to make it more succinct and clear to the readers.

6.          The article can be accepted after the corrections mentioned above.

Reviewer 2 Report

In this manuscript, LA. Basova et al. describe that MRP8/14 molecule complex may play a potential role in aggravating HIV inflammatory pathology and promoting viral replication in people with HIV (PWH) who use Meth using the U1 cell line.

It is unclear to the readers what is meant by the "HIV replication signature" in the title. Furthermore, the title does not include S100A8/A9 (MRP8/14), and the title does not summarize the content of the Abstract/MS. A change of title should be considered. In the Abstract, the description of "the systems biology approach" is insufficient as methods, and it is unclear how the authors focused on S100A8/A9; thus, the Abstract should be rewritten.

Using the U1 cell line, they conclude MRP8/MRP14 effect, but not using primary cells. As the author cited, a component of S100A8 (MRP8) has been demonstrated as an HIV inducer using a T cell line or U1 cells. In contrast, S100A8 and S100A9 are shown as HIV inhibitor factors using primary cells, and  S100A8/A9 has no impact on HIV primary macrophages and T cells (Embo J. 40: 3106540, AIDS Res Human Retro, 38: 401). The authors should perform a comparative study to demonstrate the physiological relevance of the MRP8/14 complex.  

Many publications indicate that MRP8/MRP14 is a biomarker for SARS-COV2 infection. The virus infection dramatically increases S100A8/S100A9 concentration in blood. If the authors report that S100A8/S100A9 activates latently infected cells in HIV-infected patients, co-infection of COVID and HIV would be risky. They need to discuss the impact of the complex in co-infection.

The authors could not detect soluble S1008/A9 in culture media but found it on the cell surface of the U1 cell by FACS. To confirm the binding, the authors should demonstrate a result of WB using cell membrane fraction with anti-S100A8/A9 antibody. The result would convince readers that the protein is not released but expresses on the cell surface.

How do the authors determine the concentration of S100A8/A9 in their assay even though they could not detect the protein as a soluble form?  

There are many typos in the text. The X-axis of Figure 6A contains a typo, Figure 6C needs to revise, and Fig8A needs to be provided; thus, the reviewer cannot fairly review the entire manuscript. Figure 7 font size is too small to read, and some figures are fuzzy. They need to replace them. The results of Statistical analysis need to be consistently demonstrated. The authors need to fix all of them.

S100A8/A9 is also known as calprotectin; however, the authors did not mention it. They should discuss it in the introduction.